# Combine synergetic approach with multi-scale feature fusion for Boosting Abdominal Multi-Organ and Pan-Cancer Segmentation

Shuo Wang[1][0009−0003−3022−7900] and Yanjun Peng[1][0000−0002−8444−0622]

College of Computer Science and Engineering,Shandong University of Science and Technology,Qingdao,266590, China
{pengyanjuncn}@163.com

**Abstract.** Due to the capability of abdominal images to accurately represent the spatial distribution and size relationships of lesion components in the body, precise segmentation of these images can significantly assist doctors in diagnosing illnesses. To address issues such as high computational resource consumption and inaccurate boundary delineation, we propose a two-stage segmentation framework with multi-scale feature fusion. This approach aims to enhance segmentation accuracy while reducing computational complexity. In the initial stage, a coarse segmentation network is employed to identify the location of segmentation targets with minimal computational overhead.Subsequently, in the second stage, we introduce a multi-scale feature fusion module that incorporates cross-layer connectivity. This method enhances the network's context-awareness capabilities and improves its ability to capture boundary information of intricate medical structures. Our proposed method has achieved notable results, with an average Dice Similarity Coefficient (DSC) score of 85.60% and 37.26% for organs and lesions, respectively, on the validation set. Additionally, the average running time and area under the GPU memory-time curve are reported as 11 seconds and 24,858.1 megabytes, demonstrating the efficiency and effectiveness of our approach in both accuracy and resource utilization.

**Keywords:** Deep learning · Abdominal organ segmentation · Feature fusion · Tumor segmentation

## 1 Introduction

Cancers affecting abdominal organs are a significant medical concern, particularly with colorectal and pancreatic malignancies ranking as the second and third leading causes of cancer-related mortality [6]. Computed Tomography (CT) scanning plays a crucial role in providing prognostic insights for oncological patients and remains a widely used technique for therapeutic monitoring. In both clinical research trials and routine medical practice, the assessment of tumor dimensions [3] and organ characteristics on CT scans often relies on manual two-dimensional measurements, following criteria such as the Response Evaluation Criteria In Solid Tumors (RECIST) guidelines [25]. However, this method

of evaluation introduces inherent subjectivity and is susceptible to significant inter and intra-professional variations. Furthermore, existing challenges tend to focus predominantly on specific tumor categories, such as hepatic or renal malignancies.

Convolutional neural networks (CNNs) [1] possess the capability to autonomously acquire image features by conducting convolution operations, thereby facilitating automated feature extraction. Yuan et al.[27] proposed a two-branch UNet architecture, adding a branch to the original network to learn global features.The 3D-based coarse-to-fine framework [32] enables the gradual processing of input data at various granularity levels, progressively enhancing segmentation results while conserving computational resources.Yuan et al.[28] designed a better combination of convolutional neural network and Transformer to capture dual attention features. Complementary features were generated in the Transformer and CNN domains. Feature fusion is crucial in medical image segmentation, as it integrates various pieces of information, addresses image complexity, and enhances model accuracy and generalization. UNet++[31] improved skip connections by nesting them layer and layer, and experiments on several datasets achieved perfect performance. FFA-Net [20] combines features from different levels, directing the network's attention towards more effective information. It assigns greater weight to important features while preserving shallow features.In addition, it also proposed skip connections[23] that can combine the original features while recovering the resolution. Han et al.[9] utilize deep semi-supervised learning with a precision-focused pseudo-labeling approach, effectively expanding the training dataset for liver CT image segmentation. Achieving superior results with minimal labeled data from the LiTS dataset. SS-Net[26] addresses the challenges of semi-supervised medical image segmentation by enforcing pixel-level smoothness, promoting inter-class separation, and achieving state-of-the-art performance on LA and ACDC datasets. GEPS-Net[14] combines graph-enhanced segmentation with semi-super-vised learning, notably improving pancreas segmentation on CT scans, surpassing methods with limited data, and aiding early diagnoses and adaptive therapy.

We intensity normalize and resample the size of the original image and perform extensive data enhancement. Abdominal organs as well as tumors are segmented and post-processed using a two-stage segmentation framework. The two-stage segmentation method is used to segment 3D abdominal organs and tumor images to improve accuracy, especially when dealing with complex anatomical structures, the error rate can be effectively reduced by the first stage of localization and initial segmentation, while the second stage can segment tumors and organs more finely. For large datasets, this method can reduce the computational burden and improve efficiency.

## 2   Method

Our proposed method is a whole-volume-based two-stage framework. Details about the method are described as follows:

Firstly, for the localization of organs and tumors, we adopt a lightweight model to optimize the model with fewer parameters and computational requirements; Secondly, we use mixed precision training to represent the model parameters with low accuracy, which can reduce computational overhead without significant performance loss. Finally, for duplicate inputs, cache the output results of the model to reduce duplicate calculations and improve inference speed.

### 2.1 Preprocessing

The proposed method includes the following pre-processing steps:

– Resize the image to a right-anterior-inferior (RAI) view.
– Remove the background (label 0) by threshold segmentation.
– Considering the memory constraints of the current training process, we resampled the image to a fixed size [160, 160, 160] and applied it to coarse and fine segmentation inputs.
– Intensity normalization: all images are cropped to [-500,500], and z-score normalization is applied based on the mean and standard deviation of the intensity values.
– Our framework employs a mixed-precision approach throughout the workflow to improve the efficiency of the training and testing procedures.

### 2.2 Proposed Method

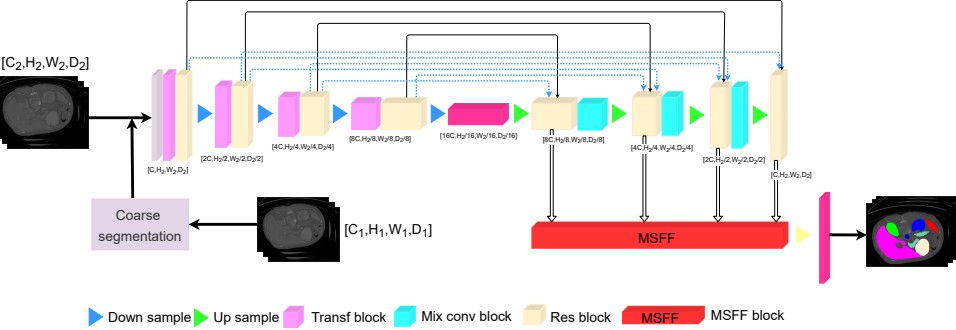

**Fig. 1.** The whole architecture of our proprosed methods. the MSFF block is the multi-scale feature fusion block, the Mixed conv block is the hybrid convolution block consisting of Conv-IN-Drop-ReLU, and the Res block represents the residual block.

The proposed network is shown in Fig.1. For abdominal medical images, the anatomical structures and lesion locations are complex and variable. The varying sizes of tumors tend to lead to category imbalance problems, and have a certain degree of artifacts and noise. To solve this issue, we design a two-stage network

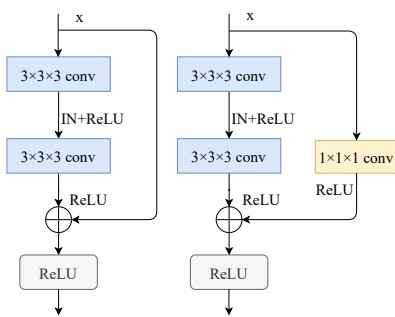

**Fig. 2.** Comparison of different residual connection methods.

[32] with multi-scale feature fusion network. We first use a lightweight U-shaped network [4] to obtain the approximate location and distribution of segmented targets. The network input is $x \in R^{B \times C \times H_1 \times W_1 \times D_1}$, where B denotes the size of the batch, C denotes the number of input channels and $H_1 \times W_1 \times D_1$ denotes the size after re-sampling. After localization, Specific optimization of the segmented target edges is performed. The network input is $x \in R^{B \times C \times H_2 \times W_2 \times D_2}$. Importantly, we design a multi-scale feature fusion module. It is used to enhance the important features in the encoding stage and improve the context-awareness of the network. It can effectively reduce the loss of information and blurred edges caused during the decoding process, thus enhancing the overall segmentation of medical images.

### 2.3   Backbone network

The two-stage framework is illustrated in Fig.1. We use a coarse segmentation network for initial localization of the segmentation target. As shown in Fig.2, a $1 \times 1 \times 1$ convolution is added to the connected path of the residuals. Compared to the original residual connection, this solves the semantic loss problem. In the deeper layers of the network, it can enhance the transfer and expression of information. It is also effective in preventing the gradient from disappearing. After pre-processing, the edges of the segmentation target are then finely segmented. Different organs or structures vary greatly in shape and size. The network channels are increased from [8,16,32,64,128] to [16,32,64,128,256] to extract richer features. This improves the ability to accurately locate details of segmented target edges.

Abdominal medical data [8] have differences in images due to differences in acquisition equipment. We combine the two residual approaches in Fig.2 to form a mixed convolution block. This block incorporates two at each layer in the encoding stage and one at each layer in the decoding stage. We use instancenorm to reinforce detailed features and enhance the consistency of intensity distribution within the region of interest. It reduces the impact of variability on feature extraction and enables better learning of the image feature representation. The

final input to the network is passed through a $1 \times 1 \times 1$ convolution to obtain a segmented probability map utilizing a sigmoid function.

Loss function: Abdominal medical images face the challenges of overlapping tissue structures and organ deformation, complicating network training. Therefore, our loss function uses a combination of binary cross entropy (BCE) loss and dice coefficient (Dice) loss [15]. It effectively solves the category imbalance problem. Our loss function expression can be described as follows:

$$L_{total} = L_{BCE} + L_{Dice} \tag{1}$$

$$L_{BCE} = -\frac{1}{N} \sum_i \sum_{c=1}^M y_{ic} \log P_{ic} \tag{2}$$

$$L_{Dice} = 1 - \frac{2|X \cap Y| + \varepsilon}{|X| + |Y| + \varepsilon} \tag{3}$$

Unlabeled data play a role in our experiments, involving the utilization of 1800 instances for inference. We divided the model training into two distinct phases, employing partially labeled data. Subsequent to model saving, predictions were applied to the entire pool of unlabeled data to generate pseudo-images and credible scores. We selected the top fifty percent most dependable instances during the prediction process. Furthermore, a new pseudo dataset is crafted by amalgamating this selection with a partially labeled dataset.

However, the outcomes didn't meet our expectations as they fell notably short. We reverted to the fully supervised approach, which yielded a 3-5% enhancement compared to previous results.

Due to time and equipment constraints, we did not use untagged images. Pseudo-labels generated by the FLARE22 winning algorithm [13] and the best-accuracy-algorithm [22] are used during the research and exploration of the methodology, and the segmentation of organs and tumors is performed using the pseudo-labeled data and the data from FLARE2023.

### 2.4   Post-processing

Utilizing the Python connected-components-3d and fastremap3 packages [30], we extract the largest connected component of the segmentation mask per each class for both coarse and fine outputs, ensuring noise impact avoidance by employing the connected component analysis and selecting the maximum connected component as the final segmentation outcome.

## 3   Experiments

### 3.1   Dataset and evaluation measures

FLARE2023 is an extension of the FLARE2021 [17] and FLARE2022 [18] challenges. This challenge aims to promote the development of universal organ

and tumor segmentation [11] in abdominal CT scans. In FLARE2023, add the lesion segmentation task. Different from existing tumor segmentation challenges [3], FLARE2023 focuses on pan-cancer segmentation, which covers various abdominal cancer types. The segmentation targets cover 13 organs and various abdominal lesions. The training dataset is curated from more than 30 medical centers under the license permission, including TCIA [5], LiTS [2], MSD [21], KiTS [10,12], autoPET [7], TotalSegmentator [24], and AbdomenCT-1K [19]. 2200 cases have partial labels and 1800 cases are unlabeled. The validation set consists of 100 CT scans of various cancer types. The test set consists of 400 CT scans of various cancer types. Specifically, the segmentation algorithm should segment 13 organs (liver, spleen, pancreas, right kidney, left kidney, stomach, gallbladder, esophagus, aorta, inferior vena cava, right adrenal gland, left adrenal gland, and duodenum) and one tumor class with all kinds of cancer types (such as liver cancer, kidney cancer, stomach cancer, pancreas cancer, colon cancer) in abdominal CT scans. All the CT scans only have image information and the center information is not available.The organ annotation process used ITK-SNAP [29], and MedSAM [16].

The evaluation metrics consist of segmentation accuracy metrics and segmentation efficiency metrics. The segmentation accuracy metricsconsist of two measures: Dice Similarity Coefficient (DSC) and Normalized Surface Dice (NSD). The segmentation efficiency metrics consist of two measures: running time (s) and area under GPU memory-time curve (MB). All measures will be used to compute the ranking. Moreover, the GPU memory consumption has a 4 GB tolerance.

### 3.2   Implementation details

**Environment settings** The development environments and requirements are presented in Table 1.

**Table 1.** Development environments and requirements.

| | |
|---|---|
| System | Ubuntu 18.04.5 LTS |
| CPU | Intel(R) Xeon(R) Silver 4210 CPU @ 2.20GHz(×8) |
| RAM | 16×4GB; 2.67MT/s |
| GPU (number and type) | NVIDIA GeForce RTX 2080Ti 11G(×4) |
| CUDA version | 11.6 |
| Programming language | Python 3.9 |
| Deep learning framework | Pytorch (Torch 1.13.0) |

**Training protocols** The training protocols of the baseline method is shown in Table 2 and Table 3

**Table 2.** Training protocols.

| Network initialization | "he" normal initialization |
|---|---|
| Batch size | 1 |
| Patch size | 160×160×160 |
| Total epochs | 200 |
| Optimizer | Adam with betas(0.9, 0.99), L2 penalty: 0.00001 |
| Initial learning rate (lr) | 0.0001 |
| Lr decay schedule | halved by 20 epochs |
| Training time | 48 hours |
| Loss function | Dice loss + BCE loss |
| Number of model parameters | 28.82M |
| Number of flops | 41.54G |

**Table 3.** Training protocols for the refine model.

| Network initialization | "he" normal initialization |
|---|---|
| Batch size | 1 |
| Patch size | 160×160×160 |
| Total epochs | 200 |
| Optimizer | Adam with betas(0.9, 0.99), L2 penalty: 0.00001 |
| Initial learning rate (lr) | 0.0001 |
| Lr decay schedule | halved by 20 epochs |
| Training time | 48 hours |
| Number of model parameters | 36.32M |
| Number of flops | 48.14G |

## 4    Results and discussion

**Table 4.** The performance on the validation set is represented by the average values in the table.

| Target | Public Validation | | Online Validation | |
|---|---|---|---|---|
| | DSC(%) | NSD(%) | DSC(%) | NSD(%) |
| Liver | 97.69±0.51 | 98.50±1.57 | 97.78 | 87.87 |
| Right Kindney | 91.74±6.79 | 91.19±8.31 | 90.32 | 84.61 |
| Spleen | 95.13±1.02 | 96.61±2.06 | 97.32 | 94.02 |
| Pancreas | 83.09±6.44 | 93.87±5.14 | 84.09 | 70.21 |
| Aorta | 94.60±1.25 | 96.84±1.98 | 91.99 | 87.59 |
| Inferior vena cava | 91.90±2.89 | 92.89±3.46 | 90.28 | 83.95 |
| Right adrenal gland | 77.96±6.91 | 90.09±2.45 | 76.46 | 80.36 |
| Left adrenal gland | 72.24±8.56 | 85.89±5.76 | 73.46 | 77.89 |
| Gallbladder | 74.02±20.45 | 73.09±24.56 | 73.73 | 62.93 |
| Esophagus | 74.41±15.67 | 85.70±19.48 | 71.31 | 62.21 |
| Stomach | 89.70±2.14 | 92.28±3.14 | 88.75 | 67.27 |
| Duodenum | 77.19±8.19 | 90.61±5.13 | 75.46 | 62.41 |
| Left kidney | 93.09±4.23 | 93.17±2.54 | 91.94 | 85.83 |
| Tumor | 37.26±23.14 | 29.09±30.41 | 39.94 | 26.47 |
| Average | 82.14±7.11 | 86.41±7.53 | 81.63 | 73.83 |

**Table 5.** Quantitative evaluation of segmentation efficiency in terms of the running them and GPU memory consumption

| Case ID | Image Size | Runnning time(s) | Max GPU(MB) | Total GPU(MB) |
|---|---|---|---|---|
| 0001 | (512,512,55) | 5.79 | 1005 | 11145 |
| 0051 | (512,512,100) | 7.12 | 1293 | 10536 |
| 0017 | (512,512,150) | 8.41 | 1940 | 10549 |
| 0019 | (512,512,215) | 10.55 | 2138 | 11474 |
| 0099 | (512,512,334) | 13.33 | 2620 | 12965 |
| 0063 | (512,512,448) | 16.81 | 2838 | 12863 |
| 0048 | (512,512,499) | 19.22 | 2985 | 13425 |
| 0029 | (512,512,554) | 23.71 | 3241 | 14562 |

### 4.1    Quantitative results on validation set

Table 4 illustrates the results of this work on the validation cases whose ground truth are publicly provided by FLARE2023. Our method performs well in the task of segmenting multiple abdominal organs. The Dice similarity coefficients (DSC) of key organs such as the liver, kidney, spleen, and aorta are all

above 0.9, and the Normalized Surface Distance (NSDs) also remain above 0.9. This highlights the superior ability of our method in capturing organ contours and morphology, proving our significant advantage in organ segmentation.

Tumor segmentation presented challenges due to uncertainties in tumor number and size, leading to recognition errors and omissions during segmentation. Consequently, the method achieved a DSC coefficient of 37.26% and an NSD coefficient of 29.09%, highlighting room for improvement in tumor recognition and delineation.Our method fully utilizes the strategy of multi-scale feature fusion, which is one of the keys to our success. By integrating image information at different scales, our model can capture the details and structures of organs more accurately. This strategy results in very satisfactory DSC and NSD values for most organs, which is a clear indication of the advantages of our method in segmentation tasks. Although we have achieved remarkable results, we recognize that there is room for further improvement in the results of tumor segmentation. Table 5 presents a quantitative evaluation of runtime and GPU memory consumption.

In our final submission, we exclusively utilized labeled data for the segmentation of abdominal organs and tumors. Our segmentation approach involved a two-stage network, which encompasses the entire segmentation process. Furthermore, we conducted ablation experiments to substantiate the benefits of employing this two-stage network. The results of our approach are presented in Table 6.

**Table 6.** Ablation research in our methodology (s represents training using a single stage network, and d represents training using a two-stage network.)

| Number | Organ DSC | Organ NSD | Tumor DSC | Tumor NSD |
|--------|-----------|-----------|-----------|-----------|
| 1(s)   | 81.40     | 79.51     | 10.25     | 9.88      |
| 2(d)   | 86.50     | 90.88     | 37.26     | 29.09     |

### 4.2 Qualitative results on validation set

In Figure 3, the upper two layers (ID13 and ID81) exhibit favorable segmentation, while the lower two layers (ID35 and ID51) display suboptimal segmentation results. The horizontal axis represents the original image, Ground Truth, ablation experiment outcomes, and segmentation results achieved through our proposed method. In instances characterized by effective segmentation, the contours of organs are distinctly delineated, highlighting the robust performance of our multi-scale method during the feature recovery phase. Conversely, for cases demonstrating inadequate segmentation, the accurate identification of organ sizes poses a challenge. Specifically, organs such as the gallbladder, duodenum, adrenal gland, and esophagus have not been precisely delineated.

Our proposed method has demonstrated effectiveness in the segmentation of multiple abdominal organs and their associated tumors. Particularly, when con-

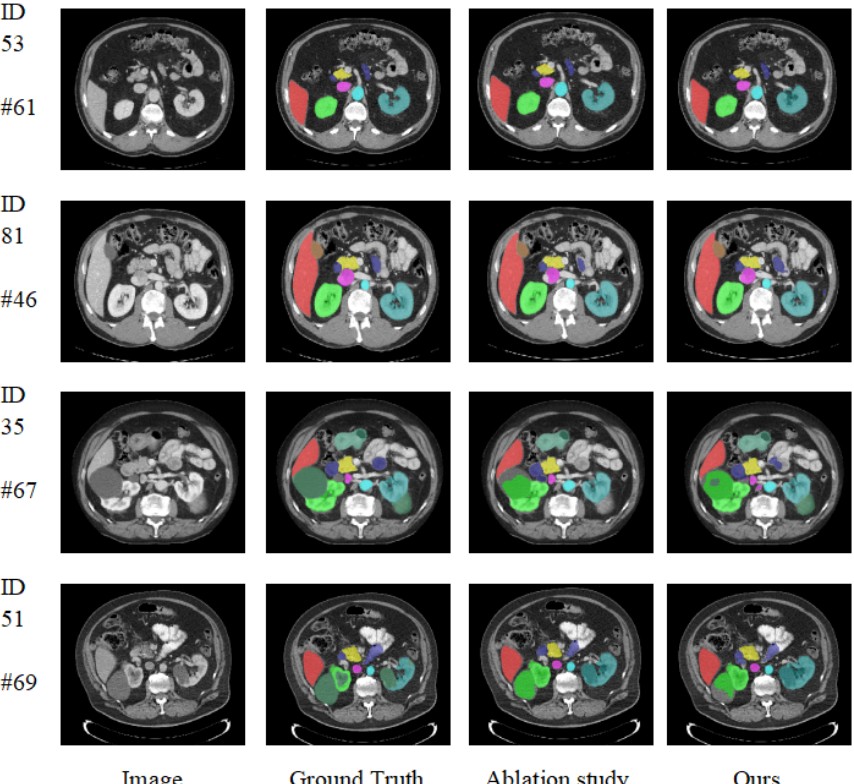

**Fig. 3.** Visualization results for some cases.

fronted with large abdominal tumors characterized by a relatively flat contour and a normal tumor count, our method exhibits high-performance segmentation, achieving notable results for both organs and tumors. Acknowledging the significance of addressing instances of segmentation failure, we delve into potential causes, including our method's limitation in accurately determining the number of tumors within the abdomen. This limitation can lead to misidentification and the overlooking of tumors. Furthermore, during the model training process, a disparity arises between tumors and organs: organs typically have fixed positions and shapes, allowing for more comprehensive feature learning, while tumors exhibit diverse positions and shapes, resulting in insufficiently learned features. To address this, we plan to enhance the model's training frequency, aiming to attain higher levels of segmentation accuracy.

### 4.3   Segmentation efficiency results on validation set

The average running time is 11.0 s per case in inference phase, and average used GPU memory is 2654 MB. The area under GPU memory-time curve is 24858.1 and the area under CPU utilization-time curve is 1240.5.

### 4.4   Limitation and future work

In our future research endeavors, we acknowledge the challenges associated with the time-consuming and labor-intensive nature of labeling medical image data for abdominal organ and tumor segmentation. Recognizing the limitations of fully supervised methods, we aim to pivot towards the advancement of semi-supervised segmentation techniques. This strategic shift involves exploring innovative approaches that effectively leverage a combination of limited annotated data and a larger pool of unlabeled data, aiming to strike a balance between accuracy and practicality in real-world medical image processing.

To address the complexities of labeling, our research will delve into the integration of advanced deep learning architectures and techniques, including self-training and consistency regularization. By harnessing the power of unlabeled data, we seek to enhance the robustness and generalization capabilities of our segmentation model. Through these efforts, our objective is to contribute significantly to the field of medical image processing, offering more accurate and efficient solutions for the segmentation of abdominal organs and tumors.

## 5   Conclusion

In this paper, our proposed network shows excellent efficacy in abdominal medical image segmentation. Through extensive experiments, we have verified the effectiveness of two-stage segmentation. Particularly, ours have achieved impressive outcomes when segmenting larger organs, and they've shown even more promising results in the context of segmenting smaller tissues. However, in the case of organ tumors, there is still a relatively long way to go.

**Acknowledgements** The authors of this paper declare that the segmentation method they implemented for participation in the FLARE 2023 challenge has not used any pre-trained models nor additional datasets other than those provided by the organizers. The proposed solution is fully automatic without any manual intervention.

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

**Table 7.** Checklist Table. Please fill out this checklist table in the answer column.

| Requirements | Answer |
| --- | --- |
| A meaningful title | Yes |
| The number of authors ($\leq$6) | 2 |
| Author affiliations and ORCID | Yes |
| Corresponding author email is presented | Yes |
| Validation scores are presented in the abstract | Yes |
| Introduction includes at least three parts: background, related work, and motivation | Yes |
| A pipeline/network figure is provided | Figure 1 |
| Pre-processing | Page 2 |
| Strategies to use the partial label | Page 4 |
| Strategies to use the unlabeled images. | Page 4 |
| Strategies to improve model inference | Page 4 |
| Post-processing | Page 5 |
| Dataset and evaluation metric section is presented | Page 5 |
| Environment setting table is provided | Table 1 |
| Training protocol table is provided | Table 2 |
| 3 | |
| Ablation study | Page number |
| Efficiency evaluation results are provided | Table 4 |
| Visualized segmentation example is provided | Figure 3 |
| Limitation and future work are presented | Yes |
| Reference format is consistent. | Yes |