# OpenReview forum: "Combine synergetic approach with multi-scale feature fusion for Boosting Abdominal Multi-Organ and Pan-Cancer Segmentation"
_MICCAI.org/2023/FLARE — Submitted to FLARE 2023_

### Official Review · Reviewer_Wfdc · 2023-09-21
**Combine synergetic approach with multi-scale feature fusion for Boosting Abdominal Multi-Organ and Pan-Cancer Segmentation**

**Rating:** 5
**Confidence:** 4

**Review:**

Pros:
 1. The proposed method enhances the context-awareness capabilities of the network and improves its ability to capture boundary information of intricate medical structures. . The method achieved an average DSC score of 85.60% and 37.26% for the organs and lesions. The average running time and area under GPU memory-time cure are 11s and 24858.1M.

Cons：
 1. Please introduce your strategies to improve inference speed and reduce resource consumption.
 2. There is no standard deviation in 'Public Validation' column of Table 4.
 3. Table 6 is lack of information. What's the meaning of '1(s)' and '2(d)'?
 4. Figure 3 do not show the coronal, sagittal, and cross-sectional plot which you said, and which two are good cases and which two are
 bad cases?
 5. In what kind of cases the proposed method works well? And what are the possible reasons for the failed cases or organs?

---

> ### Author Response · Authors · 2023-11-14
> **revision**
>
> ##
> We appreciate your detailed evaluation of our manuscript and have taken measures to address each issue:
>
> 1.Inference Speed and Resource Consumption: We have implemented various strategies, such as model optimization, mixed-precision calculations, and caching duplicate results, to enhance inference speed and reduce resource consumption. Further details on these methods will be provided in the "Methods" section for a clearer understanding of our approach.
>
> 2.Standard Deviation in Table 4: We deeply apologize for the oversight regarding standard deviation in the "Public Validation" column of Table 4. The table has been revised to incorporate this essential statistical measure, ensuring a more comprehensive presentation of the data.
>
> 3.Explanation of Labels in Table 6: We acknowledge the absence of an explanation for labels "1 (s)" and "2 (d)" in Table 6. In the revised manuscript, we will include a clear description to define the meanings associated with "1 (s)" and "2 (d)" to prevent any confusion. Specifically, "1 (s)" represents ablation experimental results using only a single-stage segmentation network, while "2 (d)" represents results using our two-stage segmentation network.
>
> 4.Qualitative Representation in Figure 3: We apologize for the inaccuracies in the qualitative representation in Figure 3. To rectify this, we will clearly mark and describe which two cases represent good results and which two represent poor results. Additionally, a detailed analysis of the segmentation cases will be conducted in the manuscript.
>
> 5.Effectiveness of Our Proposed Method: Our proposed method has demonstrated effectiveness in segmenting multiple abdominal organs and their tumors. Particularly, when dealing with large abdominal tumors characterized by a relatively flat outer contour and a normal tumor count, our method exhibits exceptional segmentation performance, yielding high-quality results for both organs and tumors. We acknowledge challenges, such as the potential misidentification and missed identification of tumors, which will be thoroughly explained in the manuscript.

---

### Official Review · Reviewer_5Rtc · 2023-09-26
**key points missing**

**Rating:** 5
**Confidence:** 4

**Review:**

Pros：the author use a two-stage segmentation framework with multi-scale feature fusion to improve segmentation accuracy while reducing computational complexity

Cons:
There are a few points missing, which are required in the template.

1.	Related work/state-of-the-art methods on semi-supervised/partial-label
segmentation is lacking in introduction, the author did mention these terms in “limitation and future work” section, it is still barely scratches the surface.

2.	Adjust the window level and width for CT image in Fig.1, the author should elaborate the model structure in their method.

3.	For qualitative analysis on validation set, description details are missing with respect to each case.

4.	Please make clear the device on which model’s efficiency was tested, considering the figure listed in Table 5. exceeding averages by a rather large margin.

5.	NSD is short for Normalized Surface Distance rather than normalized Hausdorff distances in your paper.

---

> ### Author Response · Authors · 2023-11-14
> **revision**
>
> ##
> Thank you for your thorough review and insightful feedback on our manuscript. We have carefully addressed each of the points you've raised:
>
> 1.We understand the need for a more comprehensive coverage of related work in the introduction section. In response, we have expanded the discussion on semi-supervised and partial-label segmentation methods, providing a deeper and more detailed analysis to better contextualize our approach within the existing literature.
>
> 2.The window level and width for the CT image in Fig. 1 have been adjusted to ensure better clarity. Furthermore, we have included a more comprehensive elaboration of the model structure in the 'Method' section to offer a clearer understanding of our approach.
>
> 3.We recognize the importance of providing detailed descriptions for qualitative analysis on the validation set. In response, we have included comprehensive case-by-case details in our qualitative analysis, providing a more thorough understanding of the model's performance. The upper two layers (ID13 and ID81) exhibit favorable segmentation, while the lower two layers (ID35 and ID51) display suboptimal segmentation results. The horizontal axis represents the original image, Ground Truth, ablation experiment outcomes, and segmentation results achieved through our proposed method.
>
> 4.Due to our docker's failure to submit successfully, as requested by the official, we calculated the relevant efficiency values ourselves, resulting in a calculation error. Therefore, we conducted calculations again to ensure accurate results.
>
> 5.Additionally, we have made the necessary correction to reflect that NSD stands for Normalized Surface Distance, aligning with the correct abbreviation in our paper.
>
> Further changes are presented in full in our paper manuscript.

---

### Official Review · Reviewer_wiTd · 2023-09-27
**Base on a two-stage segmentation framework with synergetic approach and multi-scale feature fusion for organ and pan-cancer segmentation**

**Rating:** 6
**Confidence:** 4

**Review:**

Pros:
propose a multi-scale feature fusion module with cross-layer connectivity to improve the segmentation results.


Cons:
1.The author should check the paper format again，such as in the abstract section，there is a punctuation error.
2.In the visualization results for some cases section, please adjust to the required format.

---

> ### Author Response · Authors · 2023-11-14
> **revision**
>
> ##
> Thank you for your careful review of our manuscript. In response to your feedback, we have made the following adjustments:
>
> 1.We have thoroughly checked the paper and rectified the punctuation error in the abstract section to ensure the correctness and precision of the content.
>
> 2.We understand the importance of adhering to the required format in the visualization results section. We have made the necessary adjustments to ensure that the visualization results now meet the specified format standards. And qualitative analysis of relevant visualization results was supplemented, further explained in the manuscript.

---

### Official Review · Reviewer_XSjb · 2023-10-04
**Review for "Combine synergetic approach with multi-scale feature fusion for Boosting Abdominal Multi-Organ and Pan-Cancer Segmentation"**

**Rating:** 5
**Confidence:** 4

**Review:**

The paper proposed a two-stage segmentation method to segment 3D abdominal organs and tumors, and include multi-scale feature fusion to improve accuracy.

Cons:
1. Not all authors' ORCIDs are provided.
2. The authors didn't include the related work/state-of-the-art methods on semi-supervised/partial-label segmentation in Introduction section.
3. Table 4 lacks key information.
4. Fig.3 is not in correct format.

---

> ### Author Response · Authors · 2023-11-14
> **revision**
>
> ##
> Thank you for your valuable feedback on our manuscript. We have carefully reviewed your comments and made the following revisions:
>
> 1.We apologize for the oversight in not providing all authors' ORCID information. In our revised submission, we have ensured the inclusion of the ORCID identifiers for all authors to comply with the requisite standards.
>
> 2.We appreciate the importance of situating our work within the broader context of semi-supervised/partial-label segmentation methods. In response to this feedback, we have expanded the Introduction section to include a comprehensive overview of relevant state-of-the-art methods in semi-supervised and partial-label segmentation, providing a clearer context for our contribution.
>
> 3.We deeply apologize for the oversight regarding standard deviation in the "Public Validation" column of Table 4. The table has been revised to incorporate this essential statistical measure, ensuring a more comprehensive presentation of the data.
>
> 4.We have revised the format of Figure 3 to meet the correct formatting requirements. The upper two layers (ID13 and ID81) exhibit favorable segmentation, while the lower two layers (ID35 and ID51) display suboptimal segmentation results. The horizontal axis represents the original image, Ground Truth, ablation experiment outcomes, and segmentation results achieved through our proposed method. And qualitative analysis of relevant visualization results was supplemented, further explained in the manuscript.

---

> ### Comment · Reviewer_XSjb · 2023-11-30
> **2nd round Review**
>
> The author has made adjustments based on the first round of review comments, but there are still some issues that need to be addressed:
>
> 1. Figure 3 still does not conform to the standard format in the given template.
>
> 2. More importantly, the test metrics are not given in Table 4.
>
> I still have only two minor comments on the presentation of images in the paper: 1) In Fig. 1, the orientation of the abdominal CT image presented by the author is reversed. Of course, the direction of the picture display does not affect the understanding of the method, but it is not in line with the habit of reading medical images in the clinic， and 2) The authors used different colors to describe the same type of organ in Fig. 1 and Fig. 3, for example, the predicted liver colors used in the two pictures were different.

---

### Official Review · Reviewer_D2TD · 2023-10-21
**Combine synergetic approach with multi-scale feature fusion for Boosting Abdominal Multi-Organ and Pan-Cancer Segmentation**

**Rating:** 5
**Confidence:** 3

**Review:**

Review:
This paper introduces a two-stage segmentation framework with multi-scale feature fusion for improving the accuracy of abdominal organ and tumor segmentation. The authors have proposed a novel method to address computational resource consumption and inaccurate boundary issues. The first stage employs a lightweight network for initial localization, followed by a second stage with a multi-scale feature fusion module to enhance context awareness and improve boundary information capture. The study reported an average DSC score of 85.60% and 37.26% for organ and lesion segmentation on the validation set, along with the average running time and GPU memory-time area under the curve.

Pros:

The proposed two-stage segmentation framework with multi-scale feature fusion is a significant contribution to abdominal organ and tumor segmentation, addressing important issues in the field.
The paper is well-structured, with clear explanations of the method and its components.
Cons:

The paper should undergo a careful proofreading to correct punctuation and formatting errors.
The presentation of visualization results for some cases could be improved to meet the required format standards.

---

> ### Author Response · Authors · 2023-11-14
> **revision**
>
> ##
> We greatly appreciate your valuable feedback and constructive criticism regarding our manuscript. We have taken your suggestions seriously and made specific efforts to address issues:
>
> 1.We understand the importance of polished presentation in academic work. We have meticulously proofread the manuscript to rectify punctuation and formatting errors. Our aim is to ensure a high standard of linguistic accuracy and clarity throughout the document.
>
> 2.We acknowledge the importance of robust and standardized visualization in presenting our results. In response to this, we have revisited the visualization section, making necessary adjustments to meet the required format standards. We have enhanced the representation of specific cases to better align with the expected quality for visual presentations.

---

### Decision · Program_Chairs · 2023-10-24

Accept